# *HLA-F* and *LILRB1* Genetic Polymorphisms Associated with Alloimmunisation in Sickle Cell Disease

**DOI:** 10.3390/ijms241713591

**Published:** 2023-09-02

**Authors:** Emmanuelle Bernit, Estelle Jean, Bastien Marlot, Laurine Laget, Caroline Izard, Isabelle Dettori, Sophie Beley, Isabelle Gautier, Imane Agouti, Coralie Frassati, Pascal Pedini, Christophe Picard, Julien Paganini, Jacques Chiaroni, Julie Di Cristofaro

**Affiliations:** 1Unité Transversale de la Drépanocytose, Centre de Référence Antilles-Guyane pour la Drépanocytose, les Thalassémies et les Maladies Constitutives du Globule Rouge et de l’Erythropoïèse, CHU Guadeloupe, 97110 Pointe à Pitre, France; 2Centre de Référence pour la Drépanocytose, les Thalassémies et les Maladies Constitutives du Globule Rouge et de l’Erythropoïèse, Assistance Publique des Hôpitaux de Marseille, 13005 Marseille, France; 3UMR7268, ADES, EFS, CNRS, Aix Marseille University, 13003 Marseille, France; 4Etablissement Français du Sang PACA Corse, 13001 Marseille, France; 5Xegen, 13420 Gemenos, France

**Keywords:** sickle cell disease, alloimmunisation, LILRB1, HLA-F, KIR3DS1, genetic polymorphisms

## Abstract

Red blood cell (RBC) transfusion remains a critical component in caring for the acute and chronic complications of sickle cell disease (SCD). Patient alloimmunisation is the main limitation of transfusion, which can worsen anaemia and lead to delayed haemolytic transfusion reaction or transfusion deadlock. Although biological risk factors have been identified for immunisation, patient alloimmunisation remains difficult to predict. We aimed to characterise genetic alloimmunisation factors to optimise the management of blood products compatible with extended antigen matching to ensure the self-sufficiency of labile blood products. Considering alloimmunisation in other clinical settings, like pregnancy and transplantation, many studies have shown that HLA Ib molecules (HLA-G, -E, and -F) are involved in tolerance mechanism; these molecules are ligands of immune effector cell receptors (LILRB1, LILRB2, and KIR3DS1). Genetic polymorphisms of these ligands and receptors have been linked to their expression levels and their influence on inflammatory and immune response modulation. Our hypothesis was that polymorphisms of *HLA Ib* genes and of their receptors are associated with alloimmunisation susceptibility in SCD patients. The alloimmunisation profile of thirty-seven adult SCD patients was analysed according to these genetic polymorphisms and transfusion history. Our results suggest that the alloimmunisation of SCD patients is linked to both *HLA-F* and *LILRB1* genetic polymorphisms located in their regulatory region and associated with their protein expression level.

## 1. Introduction

An estimated 500,000 infants are born annually worldwide with sickle cell disease (SCD); most individuals with SCD live in sub-Saharan Africa, India, the Mediterranean, and the Middle East; and approximately 100,000 individuals with SCD live in the United States [1]. SCD is a group of inherited haemoglobinopathies caused by mutations that affect the β-globin chain of haemoglobin, the most frequent of which is the *HbS* mutation [2,3,4]. Sickle haemoglobin polymerisation leading to red blood cell sickling occurs when the patient carries two mutated alleles, the most common being *HbS/HbS, HbS/HbC, HbS/HbS-β0*, or *HbS/HbS-β+* [5]; the distribution of which differs according to the geographical area [6].

The main SCD symptoms are haemolysis and vaso-occlusive crisis (VOC) [4], which may be accompanied by different clinical manifestations; the major SCD complications are acute and chronic pain, cardiopulmonary disease, central nervous system disease, and kidney disease [7]. Despite progress in treatment, the average survival of patients is about 50 years [8,9].

Red blood cell (RBC) transfusion remains a cornerstone treatment for acute and chronic severe SCD complications. RBC transfusion increases oxygen-carrying capacity, reducing anaemia, and it may also be beneficial as it increases haematocrit and reduces the synthesis of RBCs containing HbS. Transfusion also decreases HbS levels and may be beneficial in treating patients experiencing stroke, acute chest syndrome (ACS), and multiple organ failure. RBC transfusions can be provided by simple transfusion or exchange transfusions by apheresis [10]. However, the main drawbacks of RBC transfusion are iron overload, uncertain availability of compatible transfusion units, and patient alloimmunisation, which can lead to critical events such as acute or delayed haemolytic transfusion reactions (DHTR) [11,12]. DHTR can cause organ failure and death [13]. Alloimmunisation also restricts the number of compatible red blood cell units and can lead to transfusion deadlock. Some 5–75% of SCD patients develop alloantibodies, while the prevalence of RBC alloimmunisation is only 2–5% in the general population [10,13].

Anticipation of alloimmunisation is a critical issue in SCD patient care in France. RBC requirements for transfusion and their availability are imbalanced for several erythrocyte phenotypes, notably because of dramatic differences in erythrocyte antigen distribution in SCD patients and donor populations, as blood donors of African ancestry are underrepresented in Western countries. Thus, the limited availability of phenotypically matched RBCs from donors of African ancestry is an important alloimmunisation risk factor in SCD patients. To enhance the recruitment of ethnic minorities in Western countries, studies focused on barriers and motivators to blood donation have revealed significant differences between Caucasian- and African-descent respondents and have advocated dedicated campaigns [14,15].

This imbalance between supply and demand is particularly acute for the highly polymorphic RH and Kell blood group systems, for which recipient/donor compatibility is critical in transfusion practices; consequently, most alloantibodies described in SCD patients are directed against the Rh and Kell systems [10,16,17,18,19,20].

Alloimmunisation is difficult to predict, and forecasting which patients will become alloimmunised following transfusion and which will be more tolerant is not straightforward. Trend and susceptibility to generating an alloimmune response have been associated with clinical parameters and biological factors, such as the patient’s age, sex, haemoglobin mutation, or number of transfused blood products [3,12,13,21,22,23,24]; reviewed in a meta-analysis study [25].

Differences in the immune response of individuals with SCD, including pro-inflammatory status and proportion producing alloantibodies, are probably driven by genetic modifiers influencing immune regulation. Thus, research in the field of SCD alloimmunisation has naturally been focused on HLA molecules because of their key role in the presentation of foreign antigens, and this has also been made easier by immunogenetic laboratories which have paved the way in *HLA* genotyping methodologies. SCD patient alloimmunisation has been associated with HLA class II -DRB1 and -DQB1 alleles (reviewed in [26] and in [25]); however, in their meta-analysis, Wong et al. [26] were not able to confirm a significant association. Given their high level of diversity [27], the association of HLA class I (-A, -B, -C) and HLA class II (-DR, -DQ) alleles with alloimmunisation concerns a limited percentage of SCD patients. Accordingly, demonstrating that these alleles are relevant genetic factors of alloimmunisation in SCD patients requires substantial validation studies and would have a clinical benefit in providing a genetic marker for a limited percentage of the SCD population. Furthermore, to date, no functional explanation has been given in regards to how *HLA* alleles impact alloimmunisation susceptibility [26].

Other HLA molecules have been subject to clinical, genetic, and functional investigations deciphering individual differences in immune response; many studies have shown the role of the HLA Ib molecules (HLA-E, HLA-G, and HLA-F) in the immune response balance in different clinical settings where foreign or non-self antigens are exposed to recipient immune cells, such as cancer, organ transplants, or grafts. Notably, the expression of these molecules was associated with pregnancy success, which provides a practical model by which to obtain better insights into immune tolerance [28,29]. HLA Ib molecules are not involved in antigen presentation processes and do not participate in the surveillance of foreign bodies per se. These molecules are the ligands of regulatory receptors of immune effector cells (NK, T, and B lymphocytes). Their impact on immune response relies on their level of expression, in soluble form or in a microenvironmental context, which may tip the scales in favour of tolerance.

Conversely to the aforementioned HLA class I and class II genes, whose allelic diversity is a corollary of antigen presentation, the *HLA Ib* genes display a restricted allelic diversity, facilitating an understanding of the association between their genetic polymorphisms and their protein expression level and clinical relevance. HLA-E expression is strongly associated with its two main alleles [30], while polymorphisms in the regulatory regions of *HLA-G,* in linkage disequilibrium (LD) with *HLA-G* alleles defined at 8 digits, have been associated with its differential expression pattern [31,32]. The *HLA-F*01:01:02* allele (defined by rs2076183 30) was associated with higher expression in an RNAseq study [33], and three SNPs located within or near the *HLA-F* gene (rs2523393, rs1362126, and rs2523405) were shown to be associated with HLA-F mRNA and protein expression levels ([28,29,33,34,35]).

The immune tolerance functions of HLA-E and HLA-G are well described (reviewed in [36,37]), as HLA-G and HLA-E are the main ligands of inhibitory receptors expressed by immune cells.

The highest affinity of HLA-G is for the inhibitory receptors LILRB1 and LILRB2. Both receptors harbour immunoreceptor tyrosine-based inhibitory motifs (ITIMs), aiding the inhibition of intracellular signal transduction [38,39]. LILRB1 is expressed on several distinct types of immune cells, including monocytes, B cells, dendritic cells (DCs), subsets of effector and memory T cells, and 20–70% of NK cells [40,41]. Although LILRB1 displays high levels of genetic diversity, specific polymorphisms located in its regulatory region have been associated with its surface expression level on NK cells in healthy individuals: rs1004443-A, rs3760860-G, and rs3760861-G have higher levels of LILRB1 transcript and surface expression on NK cells compared with the SNPs rs1004443-G, rs3760860-A, and rs3760861-A [41]. LILRB1 expression has been associated with clinical outcome in autoimmune and inflammatory diseases and cancers, as well as with the response to bacterial and viral infections [39,42,43,44,45]. LILRB2 is involved in macrophage maturation and pro-inflammatory phenotypes, and the rs383369 SNP in a homozygous state was associated with lesser severity in inflammatory endometriosis [43], as well as in cancer response to therapy [41].

HLA-E binds to the inhibitory receptor CD94/NKG2A and to the activating receptor CD94/NKG2C, but with a weaker affinity [46,47,48]. Genes encoding for these heterodimers (*CD94, NKG2A*, and *NKG2C*) display very low, if any, genetic polymorphism [49], but the *NKG2C* gene is deleted in 20% of the worldwide population [50,51]. Therefore, some individuals will be lacking this activating receptor for HLA-E.

HLA-F seems to display a dual role: it binds to inhibitory receptors but displays its highest affinity for an activating receptor [52]. The highest affinity of HLA-F is for the activating KIR3DS1 receptor expressed on NK cells, and HLA-F binding with KIR3DS1 can trigger NK cytotoxicity and IFN-g production. Its expression has been associated with the outcome of HIV and other viral infections, cancer immune monitoring, autoimmune disease, and transplantation [53,54,55]. However, *KIR3DS1* is a common allelic variant of the *KIR3DL1* gene with an allelic frequency that varies greatly in the population (from 0 to 100%) [56,57]; therefore, individuals may be lacking this HLA-F activating receptor [53,54,55]. HLA-F can also bind to the inhibitory immune receptors LILRB1, LILRB2, and KIR3DL1 [52,53,58,59,60].

The association of the genetic polymorphisms of *HLA Ib* and of their receptors with, on one hand, their expression levels and, on the other hand, the immune response balance represents a pertinent lead in identifying genetic risk factors for SCD alloimmunisation.

The objective of this study was to help characterise patients who are likely to develop alloimmunisation and those who will be more tolerant. By identifying these groups of patients, the French Blood Centre could optimise the management of blood products by prioritising extended phenotype compatibility for patients with a higher risk of alloimmunisation.

In this study, we explored the association between genetic polymorphisms of HLA Ib ligands and of their main receptors and the risk of alloimmunisation in SCD patients who received at least one transfusion.

## 2. Results

### 2.1. Alloimmunisation of SCD Patients

Thirty-seven SCD patients were included in this study during their follow-up visit, according to inclusion/exclusion criteria defined to avoid immune-sensitive events other than those associated with SCD; thus, no patient included had experienced pregnancy, or ever had a transplant, or received a transfusion in a country other than France, or been affected by an autoimmune disease. Their biological and clinical data are described in Table 1.

Twenty-eight patients showed no antibodies directed against red blood cell antigen or HLA antigen, including the six patients who had not received any blood products. Twelve patients out of the thirty-one (38.7%) who had received blood products displayed at least one type of antibody; among them, eight patients had one antibody type, three patients had two to four antibody types, and one patient had seven antibody types (Table 2). Among the patients with antibodies directed against red blood cell antigen or HLA antigen, nine patients presented alloantibodies, and three patients displayed only one type of autoantibody (Table 2).

### 2.2. Hb Mutation Is Associated with RBC Transfusion and Alloimmunisation

The *HbS/S* genotype was more frequent in the alloimmunised patient group than in the non-alloimmunised patient group (Chi 2, *p* < 0.01). Patients with *HbS/S* or HbS/β0 received more blood products than did Sβ+ and SC patients (respective means: 124 blood products [0–696], 57.4 blood products [6–156], and 2 blood products [0–9]) (*p* = 0.01). However, no difference was observed in alloimmunised patients and non-alloimmunised patients with regards to the number of blood product transfusions (Table 1, unpaired t test, *p* = 0.28).

### 2.3. LILRB1 and HLA-F Genetic Polymorphisms Involved in Protein Expression Are Associated with SCD Patient Alloimmunisation

Missing data at a locus led to the exclusion of the concerned sample from further analyses. Sequencing of *LILRB1* and *LILRB2* genes revealed a high level of polymorphism, with 164 and 129 SNPs described within their sequence lengths of 8045 bp and 9038 bp, respectively. No identical haplotype sequence was observed twice for either gene, i.e., all individuals displayed a unique haplotype. The allelic frequencies of the non-coding polymorphisms rs3760860 and rs3760861 of *LILRB1* and rs383369 of *LILRB2* are shown in Appendix A. The rs3760860 and rs3760861 genotypes according to alloimmunisation are shown in Table 3.

The *LILRB1* non-coding polymorphisms rs3760860-A and rs3760861-A were in full linkage disequilibrium (LD), and their allelic frequencies were significantly associated with alloimmunisation (*p* = 0.02, Appendix A). When the patients’ genotypes are considered, both polymorphisms in the homozygous state displayed a trend towards significance but did not reach statistical significance (*p* = 0.05, Table 3).

The allelic frequency was higher for *LILRB2* rs383369-A in non–alloimmunised patients compared to alloimmunised patients (Appendix A, 96% vs. 75%, *p* = 0.04).

Six, four, and three alleles with allelic frequencies above 1% were characterised for the *HLA-G*, *-E*, and *-F* genes, respectively (Appendix A). The most frequent alleles were HLA-*G*01:01:01* (37.1%), *HLA-E*01:03* (51.4%), and *HLA-F*01:01:01* (77.1%).

The *HLA*-*F* non-coding polymorphism rs2523405-T and the *F*01:01:02* allele displayed an association with the absence of alloimmunisation (Table 4); for patients displaying *KIR3DS1* deletion, the differences were more pronounced and reached significance (Table 4).

No difference was observed between alloimmunised patients and non-alloimmunised patients with regards to the allelic frequency of the *HLA-G* and *HLA-E* genes (Appendix A).

*KIR3DS1* was present in 27.8% of the patients, and *NKG2C* deletion had a frequency of 31.9% (Appendix A). No association was observed between *KIR3DS1* or *NKG2C* presence and alloimmunisation (Appendix A).

Immunisation status was not associated to the *RHD/RHCE* genotype or phenotype; however, only two alloimmunised patients displayed the *RHCE*01/*01* genotype (*p* = 0.03) (Appendix A).

## 3. Discussion

RBC transfusion, the main means of therapeutic management of life-threatening SCD, is mainly limited by alloimmunisation, which can lead to major clinical complications such as haemolytic transfusion reactions and difficulties in finding future compatible transfusion units. Although some clinical and biological factors have been associated with a tendency to alloimmunisation [3,13,22,25], alloimmunisation remains difficult to predict.

In this study, we analysed the association of genetic polymorphisms of HLA Ib ligands and of their main receptors with the risk of alloimmunisation in SCD patients with no history of previous immune-sensitive events and who had received at least one RBC transfusion.

Our results support that the *LILRB1* non-coding polymorphisms rs3760860-A and rs3760861-A, in LD, are associated with alloimmunisation in SDC patients, as their allelic frequency was significantly associated with alloimmunisation. Our results suggest a dose effect as more patients displaying a homozygous genotype were alloimmunised; these results, however, did not reach statistical significance, possibly as the study size was underpowered. Our results confirm the important level of diversity of this gene [39,45], as each patient displayed a unique haplotype. Both of these polymorphisms have previously been associated with lower expression on the NK cell surface [39,44,45], inflammation and immune disorders, and cancer outcomes [42,43]. The MHC-I/LILRB1 axis is considered to be an innate immune checkpoint, and the LILRB1 inhibitory receptor participates in inflammatory and cytotoxicity response control, limiting alloreactivity [42,44,61,62,63]. Thus, our results are consistent with a lower LILRB1 surface expression on immune effector cells being associated with a weaker inhibitory signal and enhanced alloimmunisation response in SCD patients bearing rs3760860-A and rs3760861-A polymorphisms.

Our results suggesting an association of *LILRB2* rs383369-A with the absence of alloimmunisation appear to be consistent with previous results showing an association between this SNP in a homozygous state and lesser severity in inflammatory endometriosis [43], but this needs to be further investigated. No study has analysed the LILRB2 expression level according to its genotype.

Our results also support the assertion that the non-coding polymorphism of *HLA-F*, rs2523405-T, and the *F*01:01:02* allele are associated with an absence of alloimmunisation in SCD patients, especially when considering patients with no *KIR3DS1* gene. However, because of the small numbers of patients, inferences must be made with caution, even if the comparison seems statistically significant. These *HLA-F* non-coding polymorphisms were previously associated with higher levels of HLA-F expression in both immune and non-immune cell types, supporting a general mechanism of genetic regulation [28,29,34,35]. HLA-F has a dual role; it displays its highest affinity for the activating receptor KIR3DS1, which can be absent from human genomes with dramatic differences in frequency among populations, and it also binds to inhibitory receptors LILRB1, LILRB2, and KIR3DL1 [52,53,58,59,64]. Therefore, in the absence of the activating receptor KIR3DS1, HLA-F may exert an inhibitory effect on the immune system through its interaction with LILBR1, LILRB2, and KIR3DL1.

It is interesting to point out that HLA-F is expressed at low levels at the surface of resting cells and is mobilised to the cell membrane upon cell activation, such as in viral infection in immune cells; in bladder, skin, and liver carcinoma cell lines; or in inflammatory situations like severe asthma in both epithelial and immune cells [53,65,66,67,68,69,70]. SCD is associated with a chronic inflammatory status [8,9,10] where immune cells, such as neutrophils and macrophages, are activated by elevated levels of circulating haemoglobin and free heme released by haemolysis. Accordingly, patients with SCD have higher levels of pro-inflammatory cytokines, including interleukin-1, interleukin-6, and interferon-γ, compared to the levels in healthy controls [10], further supporting the hypothesis of HLA-F being expressed at the surface of different cell types and being involved in SCD immune regulation. Further investigation is required to study HLA-F cell surface expression in SCD patients compared to healthy donors and to understand its role in the immune response in SCD.

No difference was observed in alloimmunised patients and non-alloimmunised patients with regards to the allelic frequency of *HLA-G* and *HLA-E* genes. Therefore, we were unable to confirm the recent publication supporting *HLA-G* genetic polymorphisms associated with anti-Kell and anti-RH alloimmunisation in SCD patients [71].

One should consider our study as exploratory, and these results remain to be confirmed in a multicentric cohort without the very restrictive patient inclusion criteria set up here, as we chose to select patients who had not experienced any immune-sensitive events (such as pregnancy or grafts) other than those associated with SCD, thus excluding many SCD patients from follow-up. Given the small number of patients included, we only considered alloimmunisation as an endpoint, without taking into account its severity. Furthermore, historical antibodies that could not be detected in the present study were not considered, and alloimmunisation could have been underestimated in our analyses.

In conclusion, our study suggests that the genetic polymorphisms involved in the protein expression level of HLA-F and LILRB1 may influence the immune response of SCD patients following RBC transfusion. These genetic polymorphisms are easy to screen in a medical context to further investigate their clinical interest, since their analysis only requires routine molecular biology methods in immunohaematology or hospital laboratories.

Due to the complex nature of SCD regarding immune and inflammatory status, and because of the multifactorial process driving immune cell activation itself, further confirmation is needed to assess the usefulness of considering such genetic factors in SCD therapeutic management in association with other parameters such as *HbS* mutation, the number of RBC transfusion episodes, or the number of pregnancies.

In addition, the impact of *HLA-F* and *LILRB1* regulatory polymorphisms must be investigated in a broader alloimmunisation perspective, since alloimmunisation against platelets [72] or HLA [73] was reported to be associated with RBC antibodies in multiply transfused SCD patients.

## 4. Materials and Methods

### 4.1. SCD Patient Cohort and Sample Collection

SCD patients were followed up from 2007 and enrolled at the Internal Medicine Department, AP-HM, Marseille. To ensure that patient alloimmunisation was associated with transfusion, women who had been pregnant, patients transfused in a country other than France, grafted or transplanted patients, and patients with an autoimmune disease or chronic viral disease were excluded from the study. Children were not included. All participants gave their written informed consent. The protocol was approved by the Ethics Committee Sud Mediterranée IV (ID-RCB: 2017-A02744-49).

Associated clinical and biological data were collected: sex and age, haemoglobin mutation genotypes, therapeutic treatment, number of transfusions received, post-transfusion iron overload, and acute post-transfusion events (acute or delayed haemolytic transfusion reaction).

Peripheral blood was collected in a K2-EDTA tube (Diagast, Loos, France) and in a tube without anticoagulant, from which serum was collected after centrifugation at 800× *g* for 10 min. Genomic DNA was extracted from the K2-EDTA tube using a QIAcube automated process according to the manufacturer’s recommendations (Qiagen, Courtaboeuf, France).

### 4.2. Antibody Identification and Red Blood Cell Antigen Characterisation

Patients were divided into three groups: the first comprised patients who had never received a transfusion; patients who had already been transfused were then further split into two groups depending on whether they were alloimmunised or not, i.e., presenting at least one alloantibody or not. Autoantibody presence was not considered as alloimmunisation. Historical antibodies were not considered.

The detection of antibodies directed against red blood cell antigen or HLA antigen was carried out via indirect haemagglutination techniques on microplates coated with anti-IgG antiglobulin (detection of IgG antiglobulin; Qwalys Evo DIAGAST) followed by indirect haemagglutination on microfiltration supports containing polyspecific antiglobulin (Innova OCD).

Blood group phenotype analysis was performed by direct haemagglutination (microplate support: Qwalys Evo DIAGAST; microfiltration support: Innova OCD; saline tubes: DIAGAST, Loos, France) for ABO (A, B, AB), RHD/RHCE (D,C,E,c,e), KEL (Kell), FY (Fya, Fyb), JK (Jka, Jkb), and MNS (S,s).

RBC genotyping was performed using the HEA kit (Bioarray IMMUCOR^®^), which detects 24 polymorphisms associated with 38 antigens and variants. The polymorphisms analysed were CO*1/*2, DI*1/*2, DO*1/*2, HY, JO, FY*Fy, MNS*1/*2, MNS*3/*4, JK*1/*2, KEK*1/*2, KEL*3/*4 KEL*6/*7, LU*1/*2, LW*5/*7, SC*1/*2, and HBS. RHD/RHCE system genotyping was performed using RHCE and RHD kits (Bioarray IMMUCOR^®^) that detect 35 and 75 variants, respectively.

### 4.3. HLA Ib, LILRB1 and 2, KIR3DS1, and NKG2C Genetic Analyses

*HLA-E, -F,* and *-G* genotyping was performed using the HLA 11 loci NG-mix method developed in-house. Alleles were defined using HLAllele software V1 developed internally [74] using the IMGT/IPD HLA database as reference [27].

The SNPs associated with HLA-F expression, i.e., rs2523393 (*HLA-F-AS1*: intron variant, chr6:29737882, GRCh38.p14), rs1362126 (*HLA-F*: 2KB upstream variant, chr6:29723242, GRCh38.p14), and rs2523405 (*HLA-F-AS1*: intron variant, chr6: 29727528, GRCh38.p14), were sequenced by Sanger Sequencing using independent PCR (primers are described in Table 5; Taq DNA Polymerase recombinant kit, Thermo Fisher Scientific, Illkirch-Graffenstaden, France). Sequence alignment was performed in Codon Code Aligner version 10.0.2 (Codon Code Corporation) using as a reference sequence NCBI (NG_012009.1) and the IPD-IMGT/HLA database 3.37.0 [24].

The *LILRB1* (Gene ID: 10859) and *LILRB2* (Gene ID: 10288) genes were sequenced by next-generation sequencing (NGS) using a long-range PCR (Long-Range Qiagen kit) (primers are described in Table 5) and sequenced using a MiSeq NGS platform (Illumina, San Diego, CA, USA). NGS data were analysed using PolyPheMe software V.1.70 [75] (Xegen, Gémenos, France), and all polymorphic variations were screened.

The presence of *KIR3DS1* was detected using the LIFECODES KIR-SSO kit (Luminex, Immucor, Paris, France).

*NKG2C* gene deletion or presence was analysed by PCR and agarose-gel electrophoresis, using primers previously published [50] (primers are described in Table 5; Taq DNA Polymerase recombinant kit, Thermo Fisher Scientific).

### 4.4. Statistical Analysis

The primary endpoint of this study was alloimmunisation. Patients were divided into groups according to those presenting at least one alloantibody and those without any alloantibodies who had received at least one transfusion.

Allele frequencies were calculated by direct counting, and the allelic frequencies of the two groups were compared using the Chi-2 test. Quantitative variables were compared using the unpaired *t*-test or the one-way ANOVA test.

Missing data at a locus led to the exclusion of the concerned sample from further analyses; no multiple imputations were used.

Analyses were performed using GraphPad 9 software.

A *p*-value of <0.05 was used for determining whether an observed difference was statistically significant.

## Figures and Tables

**Table 1 ijms-24-13591-t001:** SCD patients’ biological and clinical history data. *p*-values comparing non-alloimmunised vs. alloimmunised patients who received at least one transfusion are given.

	Transfused Patients			Non-Transfused Patients
	Alloimmunised	Non Alloimmunised	*p*-Value	
Number of patients	9	22		6
Sex ratio (female/male)	3/6	3/19	0.20	2/4
Age (years, mean [min–max])	30 [19–53]	30 [18–61]		31 [21–48]
*HbS/HbS*	9	11	<0.01	2
*HbS/Hbβ0*	0	3	0.24	0
*HbS/Hbβ+*	0	6	0.08	0
*HbS/HbC*	0	2	0.35	4
Haemolytic transfusion reactions	5	1	<0.01	0
Transfusion iron overload	3	2	0.09	0
Treatment by hydroxyurea (%)	69.2	63.1		2
Blood product transfusion (mean number of bags [min–max])	197 [10–696]	112 [2–717]	0.28	0
Hospitalisation number/year > 3	1	3	0.85	0
Acute transfusion program	2	5	0.97	0
Number of antibodies directed against red blood cell antigen or HLA antigen (mean [min–max])	2.1 [1–7]	0		0

**Table 2 ijms-24-13591-t002:** Antibodies description (number, alloantibodies and autoantibodies).

Sample Code	Number of Antibodies	Antibodies	Allo-Antibodies	Auto-Antibodies	Transfusion (Number of Bags)
008	3	anti-Fya, anti-HLA class I, anti-Jkb	x		22
024	1	Auto anti-e		x	164
044	1	Auto anti-Jka		x	63
031	1	anti-HLA class I	x		145
009	1	Auto anti-D		x	15
023	7	Auto anti-C, Auto anti-e, anti-Fya, anti-Jkb, anti-LFA (low-frequency antigen), anti-Jka, anti-Cw, anti-Ytb	x	x	696
033	1	anti-S	x		489
005	4	Auto anti-Jkb, Auto anti-e, anti-Fy3, anti-N	x	x	10
034	1	anti-HLA class I	x		275
026	3	anti-S, anti-C, anti-E	x		20
020	1	anti-Bg1	x		108
003	1	Anti-CH/RG	x		10

**Table 3 ijms-24-13591-t003:** LILRB1 genotype according to SCD patient alloimmunisation, showing the number and frequency of patients displaying the genotype.

	Non-Alloimmunised Patients	Alloimmunised Patients	*p*-Value (Chi 2)
rs3760860-A/A or A/G and rs3760861-A/A or A/G	15/21 (71.4%)	8/8 (100%)	0.09
rs3760860-A/A and rs3760861-A/A	5/21 (23.8%)	5/8 (62.5%)	0.05

**Table 4 ijms-24-13591-t004:** HLA-F genetic polymorphisms according to patient alloimmunisation, showing the number and frequency of patients displaying the polymorphisms.

	Non-Alloimmunised Patients	Alloimmunised Patients	*p*-Value	Non-Alloimmunised Patients	Alloimmunised Patients	*p*-Value
				with no KIR3DS1 gene		
F*01:01:02	7/19 (36.8%)	1/9 (11.1%)	0.16	6/13 (46.2%)	0/6 (0%)	0.04
rs2523405-T	17/20 (85.0%)	4/8 (50%)	0.05	11/13 (84.6%)	1/5 (20%)	0.01

**Table 5 ijms-24-13591-t005:** Characteristics of primers used in this study.

Gene or SNP	Sense	Primer Sequence	PCR Product
rs1362126 (G>A) *HLA-F*: 2KB upstream variant	Forward	GAATGGGAGGCAGAAAGT	417
	Reverse	CGTGGGACTTTAGAACCT	
rs2523405 (T>G) *HLA-F-AS1*: intron variant	Forward	ATGTCCACTCGTTGCCTTTG	380
	Reverse	CACTAAACACCCAGCCCATG	
rs2523393 (A>G) *HLA-F-AS1*: intron variant	Forward	GCCATGTAAGCCAGGATGTG	506
	Reverse	ACTGTAACTGCACCTGTGGA	
*LILRB1*	Forward	AGTCTCCACATGCTCAACCA	8045
	Reverse	AGTGAGAGGGAAGGAACGTG	
*LILRB2*	Forward	CTCACCTCTGGCCTCTGTTC	9038
	Reverse	CAGGTTGCTGCAAAACTCAA	
*NKG2C*-presence	Forward	ATCAATTATTGAAATAGGATGC	363
	Reverse	CGCAAAGTTACAACCATCACCAT	
*NKG2C*-deletion	Forward	ACTCGGATTTCTATTTGATGC	411
	Reverse	ACAAGTGATGTATAAGAAAAAG	

## Data Availability

Data supporting the reported results are available upon request.

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
