# Peer review of "HLA-F and LILRB1 Genetic Polymorphisms Associated with Alloimmunisation in Sickle Cell Disease"

_ijms, 2023, doi:10.3390/ijms241713591_

Round 1

Reviewer 1 Report

The manuscript investigates the association of RBC alloimmunization with polymorphisms in multiple HLA1b proteins and their ligands. Identifying genetic factors that may predict alloimmunization prior to transfusion is a high priority and a large gap in knowledge. However, there were several factors that lessened enthusiasm for publication. 

1)  Some conclusions are not supported my statistically significant data. There are statements (below) of significant data that do not have p-values <0.05, and it is not stated in the Methods as to what p-values were used to conclude that data are significantly different.

2) Overall, the study is grossly underpowered with only 8-9 (which is unclear from the data) alloimmunized patients. Many factors analyzed may actually be significant if more patients were included in the study.

Minor comments are listed below.

Line 70: Identifying HLA class 1 and 2 alleles that predict RBC alloimmunization in SCD patients could have a dramatic clinical impact. It is unclear why the authors state that it “would have a limited clinical benefit as a genetic marker for the overall SCD population.” Please remove or re-phrase.

Line 91: Re: “Few studies have analysed LILRB2, one study associated it with less severity in inflammatory endometriosis conditions”. Does LILRB2 cause less severity, or are specific polymorphisms in LILRB2 associated with less severity?

Line 117: Re: “none had experienced pregnancy, transplant or transfusion in a country other than France, or was affected by an autoimmune disease.” None had pregnancy or transplant is France? Or none had pregnancy or transplant at all? Please clarify.

Table 1: Please define “irregular antibody”. How was it determined that an antibody was irregular?

For Alloimmunized patients, there are 9 listed. 6 are males and 6 are females? Please correct.

Please define Syklos.

How was it determined that a patient has a “non-functioning spleen”?  

Lines 129-132: HbSS and HbS/B0 patients may have higher alloimmunization frequencies because they receive more transfusions that other subtypes.

Line 140: pb should be bp

Line 147: Re: “Both polymorphisms in the homozygous state were significantly associated with higher alloimmunization (Table 2).” The p-values in Table 2 are not less that 0.05. Why do the authors say these results are statistically significant?

Table 2: The table has 8 alloimmunized patients. Table 1 has 9. Why is there a difference?

Line 151: The authors point out one significant difference in Table S2. There are other significant differences in Table S2 that are not commented upon.

Line 156: Re: “The HLA-F non-coding polymorphism rs2523405-T and the F*01:01:02 allele were significantly associated with the absence of alloimmunization (Table 3); and for patients displaying KIR3DS1 deletion, the differences were more pronounced (Table 3).” The p-values are only significant for patients with the KIR3DS1 deletion. It is unclear why the authors infer significance when the-values are not <0.05.

Table S3: There is a p-value of 0.03 for RHCE*01/*01, but no frequencies are listed for the non-alloimmunized patients. Please correct.

The associations of polymorphisms in LILRB1 with alloimmunization are not clear. There are some significant associations in Supplementary Table 2. However, the text refers to Table2 for these associations, which have p-values of 0.09 and 0.05. Clarification is needed.

Methods comments:

Were historical antibodies considered as alloimmunization or only antibodies currently showing on the screen?

What were the intervals between transfusions and antibody screening? How may antibody evanescence affect the results?

Were patients transfused with Rh/KEL matched units or extended antigen matched units?

There are some minor grammatical/spelling errors.

Author Response

To the Editor and Reviewers,

Thank you for having considered our manuscript for publication to International Journal of Molecular Sciences. The authors are grateful to reviewers for their comments and consideration. Accordingly, we are submitting a revised version of our manuscript.

Please, find in this letter responses to each point raised. The resubmission contains a Word file of the manuscript marked with yellow highlighting to indicate the changes.

*******

Reviewer #1.

The manuscript investigates the association of RBC alloimmunization with polymorphisms in multiple HLA1b proteins and their ligands. Identifying genetic factors that may predict alloimmunization prior to transfusion is a high priority and a large gap in knowledge. However, there were several factors that lessened enthusiasm for publication.

1) Some conclusions are not supported any statistically significant data. There are statements (below) of significant data that do not have p-values <0.05, and it is not stated in the Methods as to what p-values were used to conclude that data are significantly different.

The authors agree with the reviewer; the following sentence was added in the Methods section: “A p-value < 0.05 was used for determining whether an observed difference is statistically significant.”. Also, conclusions that were not supported by any statistically significant data were moderate throughout the text (abstract, results and discussion sections).

2) Overall, the study is grossly underpowered with only 8-9 (which is unclear from the data) alloimmunized patients. Many factors analyzed may actually be significant if more patients were included in the study.

The authors agree with the reviewer; results were clarified as regard to missing data.

Drawback concerning study size was underlined in the discussion as follows: “Our results support that the LILRB1 non-coding polymorphisms rs3760860-A and rs3760861-A, in LD, are associated with alloimmunization in SDC patients, as their allelic frequency was significantly associated with alloimmunization. Our results suggest a dose effect as more patient displaying homozygous genotype were alloimmunized; these results, however, did not reach statistical significance, possibly as the study size was underpowered.”; this sentence was added in the discussion: “However, because of the small numbers of patients, cautious in inferring must be taken even if the comparison seems statistically significant “ and in the conclusion” One should consider our study as exploratory, and these results remain to be confirmed in a multicentric cohort without the very restrictive patients’ inclusion criteria set up here, as we chose to select patients who had not experienced any immune sensitive events”.

Minor comments are listed below.

Line 70: Identifying HLA class 1 and 2 alleles that predict RBC alloimmunization in SCD patients could have a dramatic clinical impact. It is unclear why the authors state that it “would have a limited clinical benefit as a genetic marker for the overall SCD population.” Please remove or re-phrase.

The authors agree with the reviewer; the sentence was changed accordingly: “would have a clinical benefit as a genetic marker for a limited percentage of the SCD population.”

Line 91: Re: “Few studies have analysed LILRB2, one study associated it with less severity in inflammatory endometriosis conditions”. Does LILRB2 cause less severity, or are specific polymorphisms in LILRB2 associated with less severity?

The authors thank the reviewer; the sentence was modified as follows: “the rs383369 SNP at homozygous state was associated with less severity in inflammatory endometriosis.”

Line 117: Re: “none had experienced pregnancy, transplant or transfusion in a country other than France, or was affected by an autoimmune disease.” None had pregnancy or transplant is France? Or none had pregnancy or transplant at all? Please clarify.

The authors thank the reviewer; the sentence was modified: “no patient included had experienced pregnancy, or had a transplant at all or was transfusion in a country other than France”

Table 1: Please define “irregular antibody”. How was it determined that an antibody was irregular?

The authors agree with the reviewer, the term ‘irregular antibody” was confusing and replaced by “antibodies directed against red blood cell antigen or HLA antigen”.

For Alloimmunized patients, there are 9 listed. 6 are males and 6 are females? Please correct.

The authors thank the reviewer; the typo was corrected.

Please define Syklos.

The authors agree with the reviewer, the term Syklos was replaced by the generic name hydroxycarbamide/hydroxyurea.

How was it determined that a patient has a “non-functioning spleen”?  

The authors thank the reviewer, because non-functioning spleen or surgical splenectomy was not included as an analytical parameter, the information was deleted form the table to avoid confusing message.

Lines 129-132: HbSS and HbS/B0 patients may have higher alloimmunization frequencies because they receive more transfusions that other subtypes.

No difference was observed in alloimmunized patients and non-alloimmunized patients with regards to the number of blood product transfusions (Table 1, unpaired t test, p=0.28).

Line 140: pb should be bp

The authors thank the reviewer; the typo was corrected.

Line 147: Re: “Both polymorphisms in the homozygous state were significantly associated with higher alloimmunization (Table 2).” The p-values in Table 2 are not less that 0.05. Why do the authors say these results are statistically significant?

The authors agree with the reviewer, the sentences have been modified accordingly in the results section:” The LILRB1 non-coding polymorphisms rs3760860-A and rs3760861-A were in full Linkage Disequilibrium (LD) and their allelic frequencies were significantly associated with alloimmunization (p=0.02, Table S1). When patients’ genotypes are considered, both polymorphisms in the homozygous state displayed a trend towards significance but did not reach statistical significance (p=0.05, Table 3).”; in the discussion section the sentence was modified as follows: “Our results support that the LILRB1 non-coding polymorphisms rs3760860-A and rs3760861-A, in LD, are associated with alloimmunization in SDC patients, as their allelic frequency was significantly associated with alloimmunization. Our results suggest a dose effect as more patient displaying homozygous genotype were alloimmunized; these results, however, did not reach statistical significance, possibly as the study size was underpowered.”. This sentence was added in the conclusion “One should consider our study as exploratory, and these results remain to be confirmed in a multicentric cohort”.

Table 2: The table 2 has 8 alloimmunized patients. Table 1 has 9. Why is there a difference?

The authors agree with the reviewer. They are 9 alloimmunized patients, however, one sample could not be genotyped. The following sentence was added in the results section: “Missing data at a locus led to the exclusion of the concerned sample from further analyses.”

Line 151: The authors point out one significant difference in Table S2. There are other significant differences in Table S2 that are not commented upon.

The authors agree with the reviewer, the results presented in the Table S2 (now S1) were clarified as follows: “Allelic frequencies of the non-coding polymorphisms rs3760860 and rs3760861 of LILRB1 and rs383369 of LILRB2 are shown in Table S1. rs3760860 and rs3760861 genotype according to alloimmunization are shown in Table 3. The LILRB1 non-coding polymorphisms rs3760860-A and rs3760861-A were in full Linkage Disequilibrium (LD) and their allelic frequencies were significantly associated with alloimmunization (p=0.02, Table S1). When patients’ genotypes are considered, both polymorphisms in the homozygous state displayed a trend towards significance but did not reach statistical significance (p=0.05, Table 3).”

Line 156: Re: “The HLA-F non-coding polymorphism rs2523405-T and the F*01:01:02 allele were significantly associated with the absence of alloimmunization (Table 3); and for patients displaying KIR3DS1 deletion, the differences were more pronounced (Table 3).” The p-values are only significant for patients with the KIR3DS1 deletion. It is unclear why the authors infer significance when the-values are not <0.05.

The authors agree with the reviewer, the results section has been modified as follows: “The HLA-F non-coding polymorphism rs2523405-T and the F*01:01:02 allele displayed an association with the absence of alloimmunization (Table 4); and for patients displaying KIR3DS1 deletion, the differences were more pronounced and reached significance (Table 4).”  and the discussion has been modified as follows:” Our results also support that the non-coding polymorphism of HLA-F, rs2523405-T, and the F*01:01:02 allele were associated with absence of alloimmunization in SCD patients; especially when considering patients with no KIR3DS1 gene. However, because of the small numbers of patients, cautious in inferring must be taken even if the comparison seems statistically significant.”. Also, this sentence in the discussion was deleted: “A higher state of tolerance of SCD patients was associated with genetic polymorphisms linked to higher levels of HLA-F expression.”

Table S3: There is a p-value of 0.03 for RHCE*01/*01, but no frequencies are listed for the non-alloimmunized patients. Please correct.

The authors thank the reviewer. Missing frequencies (equal to 0%) were added to table (now S2).

The associations of polymorphisms in LILRB1 with alloimmunization are not clear. There are some significant associations in Supplementary Table 2. However, the text refers to Table2 for these associations, which have p-values of 0.09 and 0.05. Clarification is needed.

The authors agree with the reviewer, all text references to Table 3 were moderate and modified accordingly as p-values did not reach significance. The results presented in the Table S2 (now S1) were clarified as follows: “Allelic frequencies of the non-coding polymorphisms rs3760860 and rs3760861 of LILRB1 and rs383369 of LILRB2 are shown in Table S1. rs3760860 and rs3760861 genotype according to alloimmunization are shown in Table 3. The LILRB1 non-coding polymorphisms rs3760860-A and rs3760861-A were in full Linkage Disequilibrium (LD) and their allelic frequencies are significantly associated with alloimmunization (p=0.02, Table S1). When patients’ genotypes are considered, both polymorphisms in the homozygous state displayed a trend towards significance but did not reach statistical significance (p=0.05, Table 3).”

Methods comments:

Were historical antibodies considered as alloimmunization or only antibodies currently showing on the screen?

The authors agree with the reviewer. Historical antibodies were not considered in the analysis. This was specified in the Methods section.

What were the intervals between transfusions and antibody screening? How may antibody evanescence affect the results?

The authors agree with the reviewer. Most of the patient were transfused within 5 years before antibody screening, therefore, allo-immunization could have been underestimated. This drawback was added to the discussion accordingly: “Furthermore, historical antibodies that could not be detected in the present study were not considered and alloimmunization could have been underestimated in our analyses.

Were patients transfused with Rh/KEL matched units or extended antigen matched units?

Because of the great number of transfusions performed, authors cannot answer to this question.

*******

Reviewer #2.

In this small pilot study, the authors investigate HLA-F and LILRB1 genetic polymorphisms in relation to risk of red cell alloimmunization in Sickle Cell Disease patients (n=37). This small group was then further subdivided into different sickle genotypes e.g SS/SB0/SB+/SC. I have a number of concerns and comments which need to be addressed to improve suitability for publication. The most important concern I have is that the authors have over-stated the significance of their findings in this very small cohort with several subgroups. Specific points below:

1) Introduction Page 2: 'Transfusion is mainly limited by alloimmunization'. Transfusion is often limited by other factors e.g. iron overload and blood shortages in non-Western countries. 

The authors agree with the reviewer, the sentence has been modified as follows: “However, the main drawbacks of RBC transfusion are iron overload, compatible transfusion units availability and patient alloimmunization”.

Secondly, it is essential that the authors make reference to other factors which are implicated in the alloimmunization risk in SCD patients more broadly, including limited availability of non-ethnically non-phenotypically matched RCCs from African ancestry donors, even in Western countries. This has been extensively investigated by Stella Chou et al from a US perspective and factors limiting African ancestry donors from donation have been described in a European and Canadian context, respectively by Fogarty et al ' Motivators and barriers to blood donation among potential donors of African and Caucasian ethnicity' Blood Transfusion, 2023 (35848628) and Haw et al ' Sickle cell disease and the need for blood: Barriers to donation for African, Caribbean, and Black young adults in Canada' Transfusion 2023 (37194707). These should be referenced to and discussed to mention the broader societal relevance to this topic and to make this more interesting to a wider readership.

The authors agree with the reviewer, this paragraph has been added in the introduction: “RBC requirements for transfusion and their availability are imbalanced for several erythrocyte phenotypes, notably because of dramatic differences in erythrocyte antigen distribution in SCD patients and donor populations, as blood donors of African ancestry are underrepresented. Thus, limited availability of phenotypically matched RBCs from African ancestry donors is an important alloimmunization risk factor in SCD patients. To enhance recruitment of ethnic minorities in western countries, studies focused on barriers and motivators to blood donation revealed significant differences between Caucasian and African-descents respondents and advocated dedicated campaigns”.

2) The demographic details provided are insufficient. Why were these 37 patients on transfusion? Were these regular transfusions on a transfusion programme? If so, for what indication e.g. primary stroke prevention? If on regular transfusions, at what age and/or for how many years has the patient been transfused? Or how many were just intermittent transfusions e.g. in the context of an acute chest crisis. This is important as the literature suggests that transfusions during an acute crisis carry higher risk of alloimunization than planned/regular transfusions. 

The authors agree with the reviewer, demographic details were provided in Table 1 concerning hospitalization, acute transfusion program and adverse events as well as comparison between non-alloimmunized and alloimmunized patients.

3) Re Table 1 Please give the HbS/S and other genotypes in % in the table as it is unclear what Fq means and the authors later use %s to describe the distribution of these genotypes, which seems at odds with the table.

The authors agree with the reviewer, Hb genotype was corrected and numbers were given.

Don't use the term Syklos, the generic name hydroxycarbamide/hydroxyurea should be used instead.

The authors agree with the reviewer, term Syklos was replaced by the generic name hydroxycarbamide/hydroxyurea.

If some patients are on regular transfusions why are they also on Hydroxyurea? Were transfusions commenced for Hydroxyurea failure?

The authors thank the reviewer, patients were either transfused before Hydroxyurea or because Hydroxyurea failure. As immunization was not associated with Hydroxyurea treatment in this cohort, this point was not raised in the discussion.

I don't understand what the authors mean by splenectomy or non-functional spleen, as we know that all SCD patients are hyposlpenic by adulthood? If they wish to denote those who had a surgical splenectomy they should define this separately.

The authors agree with the reviewer, because surgical splenectomy was not included as an analytical parameter, the information was deleted form the table to avoid confusing message.

The term 'irregular' antibodies is insufficient in a cohort as small as this.

The authors agree with the reviewer, the term ‘irregular antibody” was confusing and replaced by “antibodies directed against red blood cell antigen or HLA antigen”.

4) Please include further details regarding the specificities of the antibodies detected e.g. Rh D/C/c/E/e, K, Fya/Fyb etc in the main manuscript text not just hidden in the Supplementary.

The authors agree with the reviewer, the Table S1 has been moved into the main manuscript as Table 2.

Generally Rh are understood to be the most common due to variant D antigens in African patients/donors which would make Rh phenotype matching more relevant than HLA typing. Either way it is still an important factor to consider and this information should be made readily available. I don't understand the nomenclature used in the Supplementary antibody table e.g. RH 1 RH2, RH3, RH5 etc. Referring to these as anti-D, anti-C etc would be more useful for the reader. It would be also useful to know after how many units transfused did these antibodies develop for each patient. The cohort is not very large so this kind of granularity should be feasible.

The authors agree with the reviewer, modifications have been made accordingly: nomenclature has been changed in the table (now) 2 and units transfused were specified for each patients.

5) I suggest a separate 2nd 'Laboratory' table for the patients baseline haematology parameters including FBC parameters like Hb, MCV, WCC, LDH, Bilirubin, Hb S%, Hb F %, Hb A%. This may help identify if those with a more haemolytic clinical phenotype are also more at risk of alloimmunization.

The author thanks the reviewer, patients baseline haematology parameters could not be retrieved, however, post transfusion events were added (HTR and transfusion iron overload) in Table 1.

6) In addition there should be comparisons between the alloimmunized and non-alloimmunized patients (by t-test/Mann Whitney or other appropriate test) for each of the parameters listed in the table 1 and the laboratory table to identify any other differences between these groups at baseline, outside of their HLA differences.

The authors agree with the reviewer, comparison between the alloimmunized and non-alloimmunized patients who received at least a transfusion for the parameters listed in the table 1 was added.

7) The LILRB1 polymorphisms described as 'associated to alloimmunization' but in fact the p-value is not <0.05 and therefore not statistically significant. The authors must point out that 1) there is a trend towards significance but that this did not reach statistical significance and 2) this may be as the study size was underpowered to show a significant difference (n=37 patients).

The authors agree with the reviewer, the sentences have been modified accordingly in the results section:” The LILRB1 non-coding polymorphisms rs3760860-A and rs3760861-A were in full Linkage Disequilibrium (LD) and their allelic frequencies were significantly associated with alloimmunization (p=0.02, Table S1). When patients’ genotypes are considered, both polymorphisms in the homozygous state displayed a trend towards significance but did not reach statistical significance (p=0.05, Table 3).”; in the discussion section as follows: “Our results support that the LILRB1 non-coding polymorphisms rs3760860-A and rs3760861-A, in LD, are associated with alloimmunization in SDC patients, as their allelic frequency was significantly associated with alloimmunization. Our results suggest a dose effect as more patient displaying homozygous genotype were alloimmunized; these results, however, did not reach statistical significance, possibly as the study size was underpowered”. This sentence was added in the conclusion “One should consider our study as exploratory, and these results remain to be confirmed in a multicentric cohort”.

8) Similarly for HAL G*01, because it is only referring to a subgroup (n=19) without KIR3DS1, this is an even smaller group and the statistically significant difference is actually based on having 0/6 patients in the allo-immunized group. I feel that with such small numbers very little can be inferred here even if the comparison seems statistically significant. These limitations must be pointed out in the discussion also and conclusions tempered by being both under-powered and not statistically significant for some associations.

The authors agree with the reviewer, the results section has been modifed as follows: The HLA-F non-coding polymorphism rs2523405-T and the F*01:01:02 allele displayed an association with the absence of alloimmunization (Table 4); and for patients displaying KIR3DS1 deletion, the differences were more pronounced and reached significance (Table 4).”  and the discussion has been modified as follows:” Our results also support that the non-coding polymorphism of HLA-F, rs2523405-T, and the F*01:01:02 allele were associated with absence of alloimmunization in SCD patients; especially when considering patients with no KIR3DS1 gene. However, because of the small numbers of patients, cautious in inferring must be taken even if the comparison seems statistically significant.”. Also, this sentence in the discussion was deleted: “A higher state of tolerance of SCD patients was associated with genetic polymorphisms linked to higher levels of HLA-F expression.”

*******

The authors would like to thank both reviewers for their careful review that have improved our manuscript. We look forward to the editor and reviewer’s comments and the editorial board’s decision.

Please do not hesitate to contact us if you require any further information.

Yours sincerely,

Reviewer 2 Report

In this small pilot study, the authors investigate HLA-F and LILRB1 genetic polymorphisms in relation to risk of red cell alloimmunization in Sickle Cell Disease patients (n=37). This small group was then further subdivided into different sickle genotypes e.g SS/SB0/SB+/SC. I have a number of concerns and comments which need to be addressed to improve suitability for publication. The most important concern I have is that the authors have over-stated the significance of their findings in this very small cohort with several subgroups. Specific points below:

1) Introduction Page 2: 'Transfusion is mainly limited by alloimmunization'. Transfusion is often limited by other factors e.g. iron overload and blood shortages in non-Western countries. 

Secondly, it is essential that the authors make reference to other factors which are implicated in the alloimmunization risk in SCD patients more broadly, including limited availability of non-ethnically non-phenotypically matched RCCs from African ancestry donors, even in Western countries. This has been extensively investigated by Stella Chou et al from a US perspective and factors limiting African ancestry donors from donation have been described in a European and Canadian context, respectively by Fogarty et al '  Motivators and barriers to blood donation among potential donors of African and Caucasian ethnicity' Blood Transfusion, 2023 and Haw et al ' Sickle cell disease and the need for blood: Barriers to donation for African, Caribbean, and Black young adults in Canada' Transfusion 2023. These should be referenced to and discussed to mention the broader societal relevance to this topic and to make this more interesting to a wider readership.

2) The demographic details provided are insufficient. Why were these 37 patients on transfusion? Were these regular transfusions on a transfusion programme? If so, for what indication e.g. primary stroke prevention? If on regular transfusions, at what age and/or for how many years has the patient been transfused? Or how many were just intermittent transfusions e.g. in the context of an acute chest crisis. This is important as the literature suggests that transfusions during an acute crisis carry higher risk of alloimunization than planned/regular transfusions. 

3) Re Table 1 Please give the HbS/S and other genotypes in % in the table as it is unclear what Fq means and the authors later use %s to describe the distribution of these genotypes, which seems at odds with the table. Don't use the term Syklos, the generic name hydroxycarbamide/hydroxyurea should be used instead. If some patients are on regular transfusions why are they also on Hydroxyurea? Were transfusions commenced for Hydroxyurea failure? I don't understand what the authors mean by splenectomy or non-functional spleen, as we know that all SCD patients are hyposlpenic by adulthood? If they wish to denote those who had a surgical splenectomy they should define this separately. The term 'irregular' antibodies is insufficient in a cohort as small as this. 

4) Please include further details regarding the specificities of the antibodies detected e.g. Rh D/C/c/E/e, K, Fya/Fyb etc in the main manuscript text not just hidden in the Supplementary. Generally Rh are understood to be the most common due to variant D antigens in African patients/donors which would make Rh phenotype matching more relevant than HLA typing. Either way it is still an important factor to consider and this information should be made readily available. I don't understand the nomenclature used in the Supplementary antibody table e.g. RH 1 RH2, RH3, RH5 etc. Referring to these as anti-D, anti-C etc would be more useful for the reader. It would be also useful to know after how many units transfused did these antibodies develop for each patient. The cohort is not very large so this kind of granularity should be feasible.

5) I suggest a separate 2nd 'Laboratory' table for the patients baseline haematology parameters including FBC parameters like Hb, MCV, WCC, LDH, Bilirubin, Hb S%, Hb F %, Hb A%. This may help identify if those with a more haemolytic clinical phenotype are also more at risk of alloimmunization. 

6) In addition there should be comparisons between the alloimmunized and non-alloimmunized patients (by t-test/Mann Whitney or other appropriate test) for each of the parameters listed in the table 1 and the laboratory table to identify any other differences between these groups at baseline, outside of their HLA differences.

7) The LILRB1 polymorphisms described as 'associated to alloimmunization' but in fact the p-value is not <0.05 and therefore not statistically significant. The authors must point out that 1) there is a trend towards significance but that this did not reach statistical significance and 2) this may be as the study size was underpowered to show a significant difference (n=37 patients).

8) Similarly for HAL G*01, because it is only referring to a subgroup (n=19) without KIR3DS1, this is an even smaller group and the statistically significant difference is actually based on having 0/6 patients in the allo-immunized group. I feel that with such small numbers very little can be inferred here even if the comparison seems statistically significant. These limitations must be pointed out in the discussion also and conclusions tempered by being both under-powered and not statistically significant for some associations.

Author Response

Marseille, 25th August 2023

Ref: ijms-2552264

Title: HLA-F and LILRB1 genetic polymorphisms associated with alloimmunization in Sickle Cell Disease

To the Editor and Reviewers,

Thank you for having considered our manuscript for publication to International Journal of Molecular Sciences. The authors are grateful to reviewers for their comments and consideration. Accordingly, we are submitting a revised version of our manuscript.

Please, find in this letter responses to each point raised. The resubmission contains a Word file of the manuscript marked with yellow highlighting to indicate the changes.

*******

Reviewer #1.

The manuscript investigates the association of RBC alloimmunization with polymorphisms in multiple HLA1b proteins and their ligands. Identifying genetic factors that may predict alloimmunization prior to transfusion is a high priority and a large gap in knowledge. However, there were several factors that lessened enthusiasm for publication.

1) Some conclusions are not supported any statistically significant data. There are statements (below) of significant data that do not have p-values <0.05, and it is not stated in the Methods as to what p-values were used to conclude that data are significantly different.

The authors agree with the reviewer; the following sentence was added in the Methods section: “A p-value < 0.05 was used for determining whether an observed difference is statistically significant.”. Also, conclusions that were not supported by any statistically significant data were moderate throughout the text (abstract, results and discussion sections).

2) Overall, the study is grossly underpowered with only 8-9 (which is unclear from the data) alloimmunized patients. Many factors analyzed may actually be significant if more patients were included in the study.

The authors agree with the reviewer; results were clarified as regard to missing data.

Drawback concerning study size was underlined in the discussion as follows: “Our results support that the LILRB1 non-coding polymorphisms rs3760860-A and rs3760861-A, in LD, are associated with alloimmunization in SDC patients, as their allelic frequency was significantly associated with alloimmunization. Our results suggest a dose effect as more patient displaying homozygous genotype were alloimmunized; these results, however, did not reach statistical significance, possibly as the study size was underpowered.”; this sentence was added in the discussion: “However, because of the small numbers of patients, cautious in inferring must be taken even if the comparison seems statistically significant “ and in the conclusion” One should consider our study as exploratory, and these results remain to be confirmed in a multicentric cohort without the very restrictive patients’ inclusion criteria set up here, as we chose to select patients who had not experienced any immune sensitive events”.

Minor comments are listed below.

Line 70: Identifying HLA class 1 and 2 alleles that predict RBC alloimmunization in SCD patients could have a dramatic clinical impact. It is unclear why the authors state that it “would have a limited clinical benefit as a genetic marker for the overall SCD population.” Please remove or re-phrase.

The authors agree with the reviewer; the sentence was changed accordingly: “would have a clinical benefit as a genetic marker for a limited percentage of the SCD population.”

Line 91: Re: “Few studies have analysed LILRB2, one study associated it with less severity in inflammatory endometriosis conditions”. Does LILRB2 cause less severity, or are specific polymorphisms in LILRB2 associated with less severity?

The authors thank the reviewer; the sentence was modified as follows: “the rs383369 SNP at homozygous state was associated with less severity in inflammatory endometriosis.”

Line 117: Re: “none had experienced pregnancy, transplant or transfusion in a country other than France, or was affected by an autoimmune disease.” None had pregnancy or transplant is France? Or none had pregnancy or transplant at all? Please clarify.

The authors thank the reviewer; the sentence was modified: “no patient included had experienced pregnancy, or had a transplant at all or was transfusion in a country other than France”

Table 1: Please define “irregular antibody”. How was it determined that an antibody was irregular?

The authors agree with the reviewer, the term ‘irregular antibody” was confusing and replaced by “antibodies directed against red blood cell antigen or HLA antigen”.

For Alloimmunized patients, there are 9 listed. 6 are males and 6 are females? Please correct.

The authors thank the reviewer; the typo was corrected.

Please define Syklos.

The authors agree with the reviewer, the term Syklos was replaced by the generic name hydroxycarbamide/hydroxyurea.

How was it determined that a patient has a “non-functioning spleen”?  

The authors thank the reviewer, because non-functioning spleen or surgical splenectomy was not included as an analytical parameter, the information was deleted form the table to avoid confusing message.

Lines 129-132: HbSS and HbS/B0 patients may have higher alloimmunization frequencies because they receive more transfusions that other subtypes.

No difference was observed in alloimmunized patients and non-alloimmunized patients with regards to the number of blood product transfusions (Table 1, unpaired t test, p=0.28).

Line 140: pb should be bp

The authors thank the reviewer; the typo was corrected.

Line 147: Re: “Both polymorphisms in the homozygous state were significantly associated with higher alloimmunization (Table 2).” The p-values in Table 2 are not less that 0.05. Why do the authors say these results are statistically significant?

The authors agree with the reviewer, the sentences have been modified accordingly in the results section:” The LILRB1 non-coding polymorphisms rs3760860-A and rs3760861-A were in full Linkage Disequilibrium (LD) and their allelic frequencies were significantly associated with alloimmunization (p=0.02, Table S1). When patients’ genotypes are considered, both polymorphisms in the homozygous state displayed a trend towards significance but did not reach statistical significance (p=0.05, Table 3).”; in the discussion section the sentence was modified as follows: “Our results support that the LILRB1 non-coding polymorphisms rs3760860-A and rs3760861-A, in LD, are associated with alloimmunization in SDC patients, as their allelic frequency was significantly associated with alloimmunization. Our results suggest a dose effect as more patient displaying homozygous genotype were alloimmunized; these results, however, did not reach statistical significance, possibly as the study size was underpowered.”. This sentence was added in the conclusion “One should consider our study as exploratory, and these results remain to be confirmed in a multicentric cohort”.

Table 2: The table 2 has 8 alloimmunized patients. Table 1 has 9. Why is there a difference?

The authors agree with the reviewer. They are 9 alloimmunized patients, however, one sample could not be genotyped. The following sentence was added in the results section: “Missing data at a locus led to the exclusion of the concerned sample from further analyses.”

Line 151: The authors point out one significant difference in Table S2. There are other significant differences in Table S2 that are not commented upon.

The authors agree with the reviewer, the results presented in the Table S2 (now S1) were clarified as follows: “Allelic frequencies of the non-coding polymorphisms rs3760860 and rs3760861 of LILRB1 and rs383369 of LILRB2 are shown in Table S1. rs3760860 and rs3760861 genotype according to alloimmunization are shown in Table 3. The LILRB1 non-coding polymorphisms rs3760860-A and rs3760861-A were in full Linkage Disequilibrium (LD) and their allelic frequencies were significantly associated with alloimmunization (p=0.02, Table S1). When patients’ genotypes are considered, both polymorphisms in the homozygous state displayed a trend towards significance but did not reach statistical significance (p=0.05, Table 3).”

Line 156: Re: “The HLA-F non-coding polymorphism rs2523405-T and the F*01:01:02 allele were significantly associated with the absence of alloimmunization (Table 3); and for patients displaying KIR3DS1 deletion, the differences were more pronounced (Table 3).” The p-values are only significant for patients with the KIR3DS1 deletion. It is unclear why the authors infer significance when the-values are not <0.05.

The authors agree with the reviewer, the results section has been modified as follows: “The HLA-F non-coding polymorphism rs2523405-T and the F*01:01:02 allele displayed an association with the absence of alloimmunization (Table 4); and for patients displaying KIR3DS1 deletion, the differences were more pronounced and reached significance (Table 4).”  and the discussion has been modified as follows:” Our results also support that the non-coding polymorphism of HLA-F, rs2523405-T, and the F*01:01:02 allele were associated with absence of alloimmunization in SCD patients; especially when considering patients with no KIR3DS1 gene. However, because of the small numbers of patients, cautious in inferring must be taken even if the comparison seems statistically significant.”. Also, this sentence in the discussion was deleted: “A higher state of tolerance of SCD patients was associated with genetic polymorphisms linked to higher levels of HLA-F expression.”

Table S3: There is a p-value of 0.03 for RHCE*01/*01, but no frequencies are listed for the non-alloimmunized patients. Please correct.

The authors thank the reviewer. Missing frequencies (equal to 0%) were added to table (now S2).

The associations of polymorphisms in LILRB1 with alloimmunization are not clear. There are some significant associations in Supplementary Table 2. However, the text refers to Table2 for these associations, which have p-values of 0.09 and 0.05. Clarification is needed.

The authors agree with the reviewer, all text references to Table 3 were moderate and modified accordingly as p-values did not reach significance. The results presented in the Table S2 (now S1) were clarified as follows: “Allelic frequencies of the non-coding polymorphisms rs3760860 and rs3760861 of LILRB1 and rs383369 of LILRB2 are shown in Table S1. rs3760860 and rs3760861 genotype according to alloimmunization are shown in Table 3. The LILRB1 non-coding polymorphisms rs3760860-A and rs3760861-A were in full Linkage Disequilibrium (LD) and their allelic frequencies are significantly associated with alloimmunization (p=0.02, Table S1). When patients’ genotypes are considered, both polymorphisms in the homozygous state displayed a trend towards significance but did not reach statistical significance (p=0.05, Table 3).”

Methods comments:

Were historical antibodies considered as alloimmunization or only antibodies currently showing on the screen?

The authors agree with the reviewer. Historical antibodies were not considered in the analysis. This was specified in the Methods section.

What were the intervals between transfusions and antibody screening? How may antibody evanescence affect the results?

The authors agree with the reviewer. Most of the patient were transfused within 5 years before antibody screening, therefore, allo-immunization could have been underestimated. This drawback was added to the discussion accordingly: “Furthermore, historical antibodies that could not be detected in the present study were not considered and alloimmunization could have been underestimated in our analyses.

Were patients transfused with Rh/KEL matched units or extended antigen matched units?

Because of the great number of transfusions performed, authors cannot answer to this question.

*******

Reviewer #2.

In this small pilot study, the authors investigate HLA-F and LILRB1 genetic polymorphisms in relation to risk of red cell alloimmunization in Sickle Cell Disease patients (n=37). This small group was then further subdivided into different sickle genotypes e.g SS/SB0/SB+/SC. I have a number of concerns and comments which need to be addressed to improve suitability for publication. The most important concern I have is that the authors have over-stated the significance of their findings in this very small cohort with several subgroups. Specific points below:

1) Introduction Page 2: 'Transfusion is mainly limited by alloimmunization'. Transfusion is often limited by other factors e.g. iron overload and blood shortages in non-Western countries. 

The authors agree with the reviewer, the sentence has been modified as follows: “However, the main drawbacks of RBC transfusion are iron overload, compatible transfusion units availability and patient alloimmunization”.

Secondly, it is essential that the authors make reference to other factors which are implicated in the alloimmunization risk in SCD patients more broadly, including limited availability of non-ethnically non-phenotypically matched RCCs from African ancestry donors, even in Western countries. This has been extensively investigated by Stella Chou et al from a US perspective and factors limiting African ancestry donors from donation have been described in a European and Canadian context, respectively by Fogarty et al ' Motivators and barriers to blood donation among potential donors of African and Caucasian ethnicity' Blood Transfusion, 2023 (35848628) and Haw et al ' Sickle cell disease and the need for blood: Barriers to donation for African, Caribbean, and Black young adults in Canada' Transfusion 2023 (37194707). These should be referenced to and discussed to mention the broader societal relevance to this topic and to make this more interesting to a wider readership.

The authors agree with the reviewer, this paragraph has been added in the introduction: “RBC requirements for transfusion and their availability are imbalanced for several erythrocyte phenotypes, notably because of dramatic differences in erythrocyte antigen distribution in SCD patients and donor populations, as blood donors of African ancestry are underrepresented. Thus, limited availability of phenotypically matched RBCs from African ancestry donors is an important alloimmunization risk factor in SCD patients. To enhance recruitment of ethnic minorities in western countries, studies focused on barriers and motivators to blood donation revealed significant differences between Caucasian and African-descents respondents and advocated dedicated campaigns”.

2) The demographic details provided are insufficient. Why were these 37 patients on transfusion? Were these regular transfusions on a transfusion programme? If so, for what indication e.g. primary stroke prevention? If on regular transfusions, at what age and/or for how many years has the patient been transfused? Or how many were just intermittent transfusions e.g. in the context of an acute chest crisis. This is important as the literature suggests that transfusions during an acute crisis carry higher risk of alloimunization than planned/regular transfusions. 

The authors agree with the reviewer, demographic details were provided in Table 1 concerning hospitalization, acute transfusion program and adverse events as well as comparison between non-alloimmunized and alloimmunized patients.

3) Re Table 1 Please give the HbS/S and other genotypes in % in the table as it is unclear what Fq means and the authors later use %s to describe the distribution of these genotypes, which seems at odds with the table.

The authors agree with the reviewer, Hb genotype was corrected and numbers were given.

Don't use the term Syklos, the generic name hydroxycarbamide/hydroxyurea should be used instead.

The authors agree with the reviewer, term Syklos was replaced by the generic name hydroxycarbamide/hydroxyurea.

If some patients are on regular transfusions why are they also on Hydroxyurea? Were transfusions commenced for Hydroxyurea failure?

The authors thank the reviewer, patients were either transfused before Hydroxyurea or because Hydroxyurea failure. As immunization was not associated with Hydroxyurea treatment in this cohort, this point was not raised in the discussion.

I don't understand what the authors mean by splenectomy or non-functional spleen, as we know that all SCD patients are hyposlpenic by adulthood? If they wish to denote those who had a surgical splenectomy they should define this separately.

The authors agree with the reviewer, because surgical splenectomy was not included as an analytical parameter, the information was deleted form the table to avoid confusing message.

The term 'irregular' antibodies is insufficient in a cohort as small as this.

The authors agree with the reviewer, the term ‘irregular antibody” was confusing and replaced by “antibodies directed against red blood cell antigen or HLA antigen”.

4) Please include further details regarding the specificities of the antibodies detected e.g. Rh D/C/c/E/e, K, Fya/Fyb etc in the main manuscript text not just hidden in the Supplementary.

The authors agree with the reviewer, the Table S1 has been moved into the main manuscript as Table 2.

Generally Rh are understood to be the most common due to variant D antigens in African patients/donors which would make Rh phenotype matching more relevant than HLA typing. Either way it is still an important factor to consider and this information should be made readily available. I don't understand the nomenclature used in the Supplementary antibody table e.g. RH 1 RH2, RH3, RH5 etc. Referring to these as anti-D, anti-C etc would be more useful for the reader. It would be also useful to know after how many units transfused did these antibodies develop for each patient. The cohort is not very large so this kind of granularity should be feasible.

The authors agree with the reviewer, modifications have been made accordingly: nomenclature has been changed in the table (now) 2 and units transfused were specified for each patients.

5) I suggest a separate 2nd 'Laboratory' table for the patients baseline haematology parameters including FBC parameters like Hb, MCV, WCC, LDH, Bilirubin, Hb S%, Hb F %, Hb A%. This may help identify if those with a more haemolytic clinical phenotype are also more at risk of alloimmunization.

The author thanks the reviewer, patients baseline haematology parameters could not be retrieved, however, post transfusion events were added (HTR and transfusion iron overload) in Table 1.

6) In addition there should be comparisons between the alloimmunized and non-alloimmunized patients (by t-test/Mann Whitney or other appropriate test) for each of the parameters listed in the table 1 and the laboratory table to identify any other differences between these groups at baseline, outside of their HLA differences.

The authors agree with the reviewer, comparison between the alloimmunized and non-alloimmunized patients who received at least a transfusion for the parameters listed in the table 1 was added.

7) The LILRB1 polymorphisms described as 'associated to alloimmunization' but in fact the p-value is not <0.05 and therefore not statistically significant. The authors must point out that 1) there is a trend towards significance but that this did not reach statistical significance and 2) this may be as the study size was underpowered to show a significant difference (n=37 patients).

The authors agree with the reviewer, the sentences have been modified accordingly in the results section:” The LILRB1 non-coding polymorphisms rs3760860-A and rs3760861-A were in full Linkage Disequilibrium (LD) and their allelic frequencies were significantly associated with alloimmunization (p=0.02, Table S1). When patients’ genotypes are considered, both polymorphisms in the homozygous state displayed a trend towards significance but did not reach statistical significance (p=0.05, Table 3).”; in the discussion section as follows: “Our results support that the LILRB1 non-coding polymorphisms rs3760860-A and rs3760861-A, in LD, are associated with alloimmunization in SDC patients, as their allelic frequency was significantly associated with alloimmunization. Our results suggest a dose effect as more patient displaying homozygous genotype were alloimmunized; these results, however, did not reach statistical significance, possibly as the study size was underpowered”. This sentence was added in the conclusion “One should consider our study as exploratory, and these results remain to be confirmed in a multicentric cohort”.

8) Similarly for HAL G*01, because it is only referring to a subgroup (n=19) without KIR3DS1, this is an even smaller group and the statistically significant difference is actually based on having 0/6 patients in the allo-immunized group. I feel that with such small numbers very little can be inferred here even if the comparison seems statistically significant. These limitations must be pointed out in the discussion also and conclusions tempered by being both under-powered and not statistically significant for some associations.

The authors agree with the reviewer, the results section has been modifed as follows: The HLA-F non-coding polymorphism rs2523405-T and the F*01:01:02 allele displayed an association with the absence of alloimmunization (Table 4); and for patients displaying KIR3DS1 deletion, the differences were more pronounced and reached significance (Table 4).”  and the discussion has been modified as follows:” Our results also support that the non-coding polymorphism of HLA-F, rs2523405-T, and the F*01:01:02 allele were associated with absence of alloimmunization in SCD patients; especially when considering patients with no KIR3DS1 gene. However, because of the small numbers of patients, cautious in inferring must be taken even if the comparison seems statistically significant.”. Also, this sentence in the discussion was deleted: “A higher state of tolerance of SCD patients was associated with genetic polymorphisms linked to higher levels of HLA-F expression.”

*******

The authors would like to thank both reviewers for their careful review that have improved our manuscript. We look forward to the editor and reviewer’s comments and the editorial board’s decision.

Please do not hesitate to contact us if you require any further information.

Yours sincerely,

Round 2

Reviewer 1 Report

Authors have responded sufficiently to all comments.

Minor edits needed

Reviewer 2 Report

The comments have been satisfactorily addressed and the manuscript accordingly improved

Minor typos